# Endometriosis and Cytoskeletal Remodeling: The Functional Role of Actin-Binding Proteins

**DOI:** 10.3390/cells14050360

**Published:** 2025-02-28

**Authors:** Wioletta Arendt, Konrad Kleszczyński, Maciej Gagat, Magdalena Izdebska

**Affiliations:** 1Department of Histology and Embryology, Faculty of Medicine, Nicolaus Copernicus University in Toruń, Collegium Medicum in Bydgoszcz, Karłowicza 24, 85-092 Bydgoszcz, Poland; warendt@cm.umk.pl (W.A.); mgagat@cm.umk.pl (M.G.); 2Department of Dermatology, University of Münster, Von-Esmarch-Str. 58, 48149 Münster, Germany; konrad.kleszczynski@ukmuenster.de; 3Faculty of Medicine, Collegium Medicum, Mazovian Academy in Płock, 08-110 Płock, Poland

**Keywords:** actin, ABPs, endometriosis, migration, adhesion, angiogenesis

## Abstract

Endometriosis is a chronic, estrogen-dependent gynecological disorder characterized by the presence of endometrial-like tissue outside the uterine cavity. Despite its prevalence and significant impact on women’s health, the underlying mechanisms driving the invasive and migratory behavior of endometriotic cells remain incompletely understood. Actin-binding proteins (ABPs) play a critical role in cytoskeletal dynamics, regulating processes such as cell migration, adhesion, and invasion, all of which are essential for the progression of endometriosis. This review aims to summarize current knowledge on the involvement of key ABPs in the development and pathophysiology of endometriosis. We discuss how these proteins influence cytoskeletal remodeling, focal adhesion formation, and interactions with the extracellular matrix, contributing to the unique mechanical properties of endometriotic cells. Furthermore, we explore the putative potential of targeting ABPs as a therapeutic strategy to mitigate the invasive phenotype of endometriotic lesions. By elucidating the role of ABPs in endometriosis, this review provides a foundation for future research and innovative treatment approaches.

## 1. Introduction

Endometriosis is a chronic inflammatory disease with significant repercussions for reproductive and overall health. According to World Health Organization (WHO), this problem affects up to 10% (190 million) of females of reproductive age [1]. In endometriosis, cells similar to the lining of the uterus are found in other locations. The primary sites of endometrial lesions are the ovaries, fallopian tubes, pelvic peritoneum, and uterosacral ligaments. Less frequent sites include the gastrointestinal tract, urinary tract, soft tissues, and areas outside the pelvic region [2]. There are several theories on how endometriosis lesions may be formed, including retrograde menstruation, cellular metaplasia, the involvement of stem cells, hematogenous or lymphatic spread, and embryogenetic theory with Müllerian rest induction [3]. Depending on the mechanism, the cells may require the ability to migrate and adapt to a new microenvironment. Available reports indicate that the epithelial–mesenchymal transition (EMT) plays a leading role here, similar to the metastasis of cancer cells [4]. Growth factors such as transforming growth factor β (TGF-β) or epithelial growth factor (EGF) induce EMT through the activation of Wnt and Notch signaling pathways. It activates the downstream transcription factors, such as Snail, zinc finger E-box binding homeobox (ZEB), suppressor of mothers against decapentaplegic (SMAD), and Twist [5]. As a result, epithelial cadherin (E-cadherin) is degraded, which leads to plasma membrane disintegration and disrupts the interactions with β-catenin. Simultaneously, the expression of the EMT markers, neural cadherin (N-cadherin) and vimentin, rise [6]. Available scientific reports indicate a correlation between endometriosis and the EMT process, showing a decrease in the expression of E-cadherin, desmoplakin, occludin, and claudin, alongside an increase in the expression of N-cadherin, vimentin, and fibronectin compared to normal endometrium [7]. In addition, changes in the activity of matrix metalloproteinases (MMPs) occur, which promotes cell migration followed by adhesion. The growth and development of endometrial cells at ectopic sites can occur through multiple pathways, including Wnt/β-catenin, (nuclear factor kappa-light-chain-enhancer of activated B cells) (NF-κB), mitogen-activated protein kinase (MAPK)/MAPK/ERK (MEK)/extracellular signal-regulated kinase (ERK), phosphoinositide 3-kinase (PI3K)/Akt/mammalian target of rapamycin (mTOR), Ras homolog family member (Rho)/Rho-associated protein kinase (ROCK), reactive oxygen species (ROS), and cytokines and the underlying mechanisms of pathophysiology include proliferation, migration, invasion, fibrosis, angiogenesis, oxidative stress, and inflammation [8]. These processes are accompanied by a rapid reorganization of the cytoskeleton, growth of migratory protrusions, and formation of stable focal contacts. Cytoskeletal rearrangement is controlled by the interaction of various actin-binding proteins (ABPs) with the actin cytoskeleton, and is crucial for cell specialization, cell–cell communication, and adhesion, maintaining cell shape, as well as in the development of specialized migratory structures (lamellipodia and filopodia) (Figure 1) [9]. Numerous reports point to the changes in the expression of various ABPs in endometriosis lesions compared to normal endometrium [10,11]. Despite multiple theories and proposed molecular pathways, the precise mechanism behind endometriosis development remains elusive. Consequently, therapeutic options are currently limited to symptomatic treatments, including pain management, hormonal therapy, and surgical resection. However, even after successful lesion removal, recurrence rates range from 6% to 67% among patients, varying based on the criteria used in the study [12].

This review aims to provide a comprehensive summary of the roles of ABPs in endometriosis. Additionally, it delves into future research directions that could enhance our understanding of endometriosis development, and identify potential therapeutic, diagnostic, and preventive targets.

To compile the relevant literature, we conducted a systematic search in the PubMed database using the keywords “[name of ABP] + endometriosis”. We included only articles published in English and focused primarily on studies conducted on human samples. Studies involving other species were excluded, except for one study on baboons and those that incorporated mouse models as complementary experimental systems alongside human data. Furthermore, publications containing unscientific claims without support in the scientific literature were also excluded.

## 2. ABPs in Endometriosis

ABPs are crucial regulators of cytoskeletal dynamics, and their dysregulation has been implicated in numerous conditions, including endometriosis. In mammals, over 50 ABPs have been identified [13], but only a subset has been linked to the formation or progression of endometriotic lesions.

This section discusses all ABPs that have been reported in the literature to play a role in the pathophysiology of endometriosis.

### 2.1. Alpha-Actinin

Alpha-actinins are part of the spectrin gene superfamily. They are ABPs with various functions across different cell types. In non-muscle cells, the cytoskeletal isoform is found along microfilaments and adherens junctions, playing a role in binding actin to membranes. Alpha-actinin-1 (ACTN1) is a protein that crosslinks filamentous actin (F-actin) and anchors it to intracellular structures. Toniyan et al. found that ACTN1 was significantly upregulated in ectopic versus eutopic endometrium [14]. These changes were confirmed at both ACTN1 protein and *ACTN1* mRNA levels, with increases of 75% and 130% compared to normal endometrium, respectively. It is worth noting that this study was a single patient case study with rare cervical endometriosis. However, similar results were shown by the same team in a larger-scale study, including 12 controls and 15 endometriosis cases with different lesion localizations (5 adenomyosis, 4 external genital endometriosis, 4 extragenital endometriosis, and 2 umbilical endometriosis). The authors confirmed that ACTN1 protein and *ACTN1* mRNA levels were elevated in all endometriosis samples compared to controls, especially in umbilical endometriosis, where the increase reached 273% and 266% for protein and mRNA levels, respectively. Interestingly, ACTN1 expression appeared to increase progressively with increasing distance from the normal endometrium [15]. These results highlight the potential of ACTN1 as a molecular target in endometriosis. The binding of ACTN1 is crucial during the formation of stress fibers and cell movement, which are processes that contribute to the ectopic implantation and persistence of endometriotic lesions [16,17]. Notably, ACTN1 has also been implicated in pathologies such as tumorigenesis and drug resistance in multiple cancer types [18,19,20]. Additionally, the activity of ACTN1 is influenced by factors such as phosphorylation of the tyrosine residue by focal adhesion kinase (FAK), and increased intracellular calcium concentration, which blocks its interactions with F-actin [21]. Currently, there are no reports on the impact of ACTN1 expression on the migratory potential of endometriosis cells, but some evidence suggests its role in the development of endometriosis. Study by Slater et al. showed a complete knockout of ACTN after immunohistochemical staining of endometriosis samples (10 patients), while a positive signal was found in normal uterine epithelium (6 patients) [10]. Similar observations were made for endometrioid cancer samples. The authors suggest that the loss of expression of ACTN and other ABPs favors tissue breakdown and facilitates cell migration. However, the specific ACTN isoform was not specified in this study. Simultaneously, studies showed no significant difference in the expression of alpha-actinin-4 (ACTN4) between ectopic and eutopic endometrium in both studies by Toniyan et al. [14,15].

### 2.2. Calponin

Calponin (CNN), a cytoskeletal protein predominantly expressed in smooth muscle cells, has also been identified in non-muscle cells such as fibroblasts, endothelial cells, and various cancer cells, where it plays a critical role in cytoskeletal dynamics, cell motility, adhesion, and migration. Recent research has highlighted its involvement in endometriosis, particularly in the stromal remodeling and smooth muscle metaplasia characteristic of this disease [22].

Studies have demonstrated that calponin-1 (CNN1) is significantly upregulated in ovarian endometriotic lesions compared to eutopic endometrium, at both *CNN1* mRNA and CNN1 protein levels. Its expression is primarily localized in the stroma of endometriotic lesions, with some variability attributed to biological and histological differences among the patients. Furthermore, CNN1 has been identified as a downstream target of the TGFβ1 signaling pathway, which is hyperactivated in ovarian endometriosis. This suggests a regulatory role of CNN1 in processes such as stromal remodeling and smooth muscle metaplasia, pivotal to the progression of endometriotic lesions [23].

Additional insights into CNN’s role have been provided by studies on superficial peritoneal endometriosis. Ibrahim et al. observed a distinct spatial distribution of CNN within lesions. Its expression was elevated in the central compartment, populated predominantly by myofibroblasts of contractile phenotype, compared to the peripheral compartment, which was primarily composed of collagen I-producing cells exhibiting a secretory phenotype. This spatial variability underscores the functional heterogeneity of stromal cells within endometriotic lesions. CNN’s presence in myofibroblasts and smooth muscle-like cells underscores its role in tissue remodeling, which may facilitate microtrauma and the implantation of retrograde endometrium, leading to new lesion formation. The study further supports the connection between metaplasia of myofibroblasts, CNN expression and TGFβ1 activity [24].

These findings suggest that CNN could help characterize cell phenotypes across different stages of myofibroblastic metaplasia. Additionally, its expression in myofibroblasts and smooth muscle-like cells highlights its involvement in tissue remodeling, a process that may contribute to microtrauma and promote the implantation of retrograde endometrial fragments, ultimately leading to the formation of new lesions. Although these findings suggest potential therapeutic targets, further research is needed to develop specific treatment strategies targeting CNN in the context of endometriosis.

### 2.3. Cofilin-1

Cofilins (CFLs) regulate actin dynamics, promoting actin branching and cytoskeleton reorganization. Cofilin-1 (CFL1) is crucial for processes like tumor progression, cell motility, adhesion, invasion, and angiogenesis. In the human uterine endometrium, CFL colocalizes with G-actin on the apical side of luminal epithelial cells during the proliferative phase, and shifts to the basolateral compartment during the secretory phase [11].

Morris et al. used a baboon model to study CFL localization in endometriosis. They found that CFL’s expression pattern in the proliferative phase was similar in both healthy and endometriosis groups. During the secretory phase, CFL translocated in healthy endometrium but not in endometriosis-affected endometrium [11]. It may explain the difficulties in embryo implantation and high infertility rates connected with endometriosis. Similar patterns were seen with slingshot (SSH1), a phosphatase regulating CFL activity, indicating new targets for endometriosis-related infertility research [11].

Xu et al. found that CFL1 is overexpressed in eutopic endometrium of patients with ovarian endometriosis, contributing to increased proliferation, adhesion, invasion, angiogenesis, and decreased apoptosis. The authors also showed that silencing *CFL1* mitigated these effects [25]. It was further supported by Chen et al., who also confirmed CFL1 upregulation in the endometrium of women with endometriosis compared to the samples of disease-free patients [26].

One of the proteins interacting with CFL1 is LIM domain kinase 1 (LIMK1). Liu et al. indicate that estradiol mediates the phosphorylation of CFL1 via LIMK1, influencing cell invasion and proliferation in endometriosis patients. They observed abnormally high expression of LIMK1 and CFL1, which was closely associated with increased invasiveness and proliferation of eutopic endometrial stromal cells from endometriosis patients compared to those from the endometrium of control group. Silencing *LIMK1* using RNAi significantly reduced estradiol-induced CFL1 phosphorylation [27]. These findings align with previous studies, demonstrating that ectopic endometrial stromal cells exhibit higher LIMK1 expression than their eutopic counterparts. Furthermore, overexpression of LIMK1 in eutopic endometrial stromal cells led to an increase in migration, invasion, proliferation, and elevated levels of adhesion, invasion, and angiogenesis markers, while LIMK1 knockdown in ectopic endometrial stromal cells resulted in the opposite effects [28]. Another protein related to CFL1 in endometriosis is platelet-derived growth factor (PDGF), which induces cell proliferation. Wang et al. found that PDGF increased *CFL1* expression in eutopic endometrium stromal cells and significantly promoted their proliferation in a time- and dose-dependent manner. Silencing *CFL1* significantly reduced PDGF’s proliferative effect, supporting the hypothesis of CFL1’s role in endometriosis-related infertility [29]. Lee et al. reported that *PDGF-A* mRNA levels were significantly lower in the eutopic endometrium of endometriosis patients than in the control group during the secretory phase [30]. These findings suggest that PDGF may influence CFL1 levels and maybe even translocation capacity, thus, also the process of embryo implantation and fertility. CFL1 activity is also influenced by testis-specific protein kinase 1 (TESK1), which phosphorylates CFL1 at Ser3, blocking its actin-severing abilities and affecting actin dynamics [31]. Yotova et al. found the long intergenic noncoding RNA 01,133 (LINC01133) knockdown in the 12Z endometriosis cell line upregulated TESK1 and increased CFL phosphorylation [32].

### 2.4. Ezrin–Radixin–Moesin Family

The ezrin–radixin–moesin (ERM) family consists of three closely related proteins: ezrin (EZR), radixin (RDX), and moesin (MSN). These proteins primarily crosslink integral plasma membrane proteins or scaffolding proteins to F-actin. Beyond their expression, the activity of ERM proteins, regulated by phosphorylation and dephosphorylation cycles, is crucial as it allows them to switch between open (active) and closed (inactive) conformations. Slater et al. reported that in normal endometrial tissue, EZR-positive regions were predominantly localized in the apical, lateral, and basal parts of glandular epithelium. In contrast, in endometriosis samples, EZR labeling was nearly absent, suggesting a complete loss of its expression. However, the study relied solely on immunohistochemical labeling without quantitative protein analysis [10]. In turn, Ornek et al. demonstrated EZR and phosphoezrin (pEZR) expression across all analyzed tissues, including normal endometrium from endometriosis-free individuals and paired eutopic and ectopic endometrium from endometriosis patients. EZR showed stronger signals in eutopic and ectopic endometrium compared to controls, primarily in glandular cells, while pEZR was mostly detected in stromal cells, with restricted localization to the apical surface in glandular epithelium. Quantitative analysis confirmed a progressive increase in EZR and pEZR levels from controls to eutopic and ectopic samples, correlating with enhanced cellular invasiveness [33]. Similarly, Chen et al. observed EZR expression in both glandular and stromal cells of ovarian lesions, with intensity progressively increasing between controls, eutopic, and ectopic endometrium. In stromal cells isolated from ectopic endometrium, inhibiting EZR phosphorylation using NSC305787 significantly reduced pEZR levels without affecting total EZR levels. This inhibition impaired the formation of actin-rich migratory protrusions, reduced F-actin levels, and decreased migratory and invasive potential [34]. NSC305787 inhibits EZR phosphorylation mediated by protein kinase C iota (PKCι), a member of the protein kinase C family of serine/threonine kinases.

Positive EZR labeling in glandular epithelium and pEZR in stromal cells was further confirmed in studies by Peloggia et al., who found no differences in EZR or pEZR expression and localization between endometriosis tissue samples from different sites or phases of the menstrual cycle [35]. Jiang et al. reported a gradual increase in *EZR* mRNA expression from control to eutopic and ectopic endometrium groups, with no significant differences between samples collected during the proliferative and secretory phases of the menstrual cycle. Additionally, siRNA-mediated downregulation of *EZR* in stromal cells isolated from ectopic endometrium significantly reduced their migratory potential compared to controls transfected with non-specific siRNA [36]. Bernacchioni et al. highlighted the association between EZR and sphingosine-1-phosphate receptor 3 (S1PR3), a G protein-coupled receptor associated with angiogenesis and endothelial function. S1PR3 expression was upregulated in endometriosis lesions compared to control endometrium and positively correlated with EMT and fibrosis markers. The application of an EZR inhibitor (NSC668394) abolished the effects of S1PR3 activation, emphasizing the importance of EZR activity in EMT. Studies using the 12Z cell line, derived from endometriotic glandular cells, confirmed that EZR activation is critical for EMT, as inhibiting EZR phosphorylation significantly reduced EMT marker expression [37]. NSC668394 is another commercially available compound that inhibits ezrin T567 phosphorylation mediated by PKCι.

Yotova et al. demonstrated that stromal cells isolated from ectopic endometrium exhibited elevated phosphorylation of all ERM proteins compared to those from eutopic endometrium and controls. This higher ERM activity was associated with a more contracted cellular morphology and reduced migratory potential, suggesting that ERM phosphorylation may anchor ectopic cells and stabilize lesions at their implantation sites. They further showed that the rapidly accelerated fibrosarcoma-1 (Raf-1)/Rho-associated coiled-coil kinase 2 (ROCKII) pathway modulates cytoskeletal reorganization and motility in endometriotic stromal cells, where low Raf-1 levels correlated with hyperactivation of ROCKII and increased ERM phosphorylation [38]. Interestingly, Moggio et al. observed contrasting results in mesenchymal cells isolated from ectopic endometrium, where a lower pEZR-to-EZR ratio following sorafenib treatment correlated with reduced migratory but not invasive properties. Sorafenib, a multi-tyrosine kinase inhibitor targeting Raf kinases, led to decreased EZR phosphorylation in mesenchymal cells from control, eutopic, and ectopic endometrium, suggesting distinct mechanisms of ERM regulation [39].

The roles of RDX and MSN in endometriosis are even less well understood. Zhang et al. showed that extracellular matrix protein 1 (ECM1) knockdown in hEM15A, an immortalized stromal cell line resembling eutopic stromal cells from women with endometriosis, significantly decreased *RDX* expression, likely through the RhoC/ROCK1 pathway, leading to reduced migratory potential. However, the lack of validation in primary stromal cells remains a limitation [40]. Ametzazurra et al. further showed that MSN levels in endometrial fluid differed significantly between disease-free controls and patients with early or advanced endometriosis, proposing its potential diagnostic value. MSN is also significantly upregulated in exosomes and vaginal secretions from endometriosis patients compared to disease-free controls [41]. Abdula et al. demonstrated that exosomes produced by endometriotic stromal cells enhanced normal stromal cell migration, increased tubular structure formation by endothelial cells, and elevated inflammatory cytokine production in ovarian epithelial cells. These effects were mediated by MSN, suggesting that exosomal MSN severely impacts surrounding normal cells [42].

Together these reports highlight that ERM proteins play a crucial role in the migration and adhesion of ectopic endometrial cells. Enhanced activation of EZR, particularly in stromal cells, is a relatively consistent observation in endometriosis tissue samples. The findings on MSN are also intriguing, as it can be transported via exosomes and influence the behavior of normal cells surrounding endometriotic lesions. Moreover, studies suggest the potential diagnostic value of ERM proteins in endometriosis.

### 2.5. Fascin

Fascin-1 (*FSCN1*) is a critical regulator of cytoskeleton that facilitates the bundling of microfilaments into tight, parallel structures essential for the formation of actin-rich migratory protrusions. These structures are pivotal for cell motility and invasion [43,44]. Upregulated *FSCN1* expression has been implicated in cancer metastasis, making it a promising therapeutic target in aggressive malignancies [45,46,47,48].

Emerging evidence highlights *FSCN1*’s role in endometriosis, particularly in driving the invasive behavior of ectopic endometrial cells. Luo et al. demonstrated that *FSCN1* is upregulated in ectopic endometrium compared to eutopic samples. Treatment of endometriotic cells with an autophagy activator suppressed proliferation, migration, and invasiveness, accompanied by shorter actin-rich protrusions. However, exogenous upregulation of *FSCN1* reversed these effects, confirming its integral role in filopodia formation and its association with enhanced cellular invasiveness in endometriotic lesions [49].

Further studies by Adammek et al. corroborate these findings, demonstrating that *FSCN1* significantly influences cytoskeletal organization and cellular motility in endometriotic cells. Overexpression of miR-145, a regulatory microRNA, resulted in a significant downregulation of *FSCN1* at the mRNA level, leading to decreased invasiveness and motility in both the 12Z endometriotic cell line and primary stromal cells derived from patients. Alongside *FSCN1*, miR-145 modulates a network of other targets, including pluripotency markers (SRY-box transcription factor 2 (SOX2), Kruppel-like factor 4 (KLF4)) and cell adhesion molecules (e.g., junctional adhesion molecule A (JAM-A)), which collectively contribute to the growth, invasiveness, and persistence of endometriotic lesions [50]. Similarly, Wang et al. identified *FSCN1* as a direct target of miR-145, reinforcing its regulatory role in endometriosis pathogenesis [51].

These findings highlight *FSCN1* as a pivotal mediator of the invasive phenotype in endometriosis, associating its expression with cytoskeletal dynamics, cell motility, and the persistence of ectopic lesions. Targeting *FSCN1* or its regulatory pathways, such as miR-145 modulation, offers a promising avenue for therapeutic intervention in endometriosis.

### 2.6. Myosins and Caldesmon

Myosins (MYOs) are a family of actin-binding motor proteins that play critical roles in various cellular processes, including intracellular transport, cell division, adhesion, and motility. In muscle cells, MYOs are essential for contraction, forming the thick filaments of the sarcomere and generating force through cyclic interactions with actin filaments powered by adenosine triphosphate (ATP) hydrolysis. In non-muscle cells, they contribute to cytoskeletal organization and dynamic cellular changes, processes integral to tissue remodeling [52].

In the context of endometriosis, MYOs play a significant role due to their involvement in cellular dynamics within endometriotic lesions. These lesions often contain not only glandular epithelial and stromal cells but also populations of myofibroblasts and smooth muscle-like cells, which express smooth muscle markers such as smooth muscle myosin heavy chain (SM-MHC), desmin, smooth muscle actin (SMA), and caldesmon (CALD). These markers highlight the contractile and fibrotic features of endometriotic lesions and their involvement in pathological tissue remodeling [22].

Barcena de Arellano et al. demonstrated that smooth muscle-like cells associated with endometriosis exhibit markers indicative of contractile capabilities, including SM-MHC, SMA, desmin, and CALD [53]. These cells also express receptors for vasopressin and oxytocin, suggesting their potential to contract in response to hormonal stimulation. Such contractions could activate nociceptors in the peritoneum, contributing to pelvic pain, a hallmark symptom of endometriosis [53,54,55]. This finding underscores the potential of targeting smooth muscle contractility as a therapeutic strategy to alleviate pain. Similarly, Wang et al. confirmed strong positive signals for SM-MHC, desmin, and α-SMA in ovarian endometriosis tissue samples [56]. Additionally, van Kaam et al. demonstrated that deeply infiltrating endometriosis lesions are characterized by fibromuscular tissue containing myofibroblasts expressing smooth muscle markers such as α-SMA, desmin, and SM-MHC [57]. Their findings support the idea that smooth muscle-like cells in deeply infiltrating lesions primarily arise from the transdifferentiation of local fibroblasts in response to ectopic endometrial tissue, rather than from metaplasia of the endometrial cells themselves. A nude mouse model provided further evidence, showing α-SMA expression induced in host fibroblasts surrounding human endometrial implants. This process mimics stromal reactions observed in wound healing and fibrocontractive diseases, characterized by extracellular matrix deposition and collagen production.

The genetic aspects of MYO involvement in endometriosis have also been explored. Lou et al. identified *MYH8* missense mutations in two out of 152 genomic DNA samples from Han Chinese women with ovarian endometriosis [58]. These mutations were absent in controls and predicted to be pathogenic. However, their clinical significance remains unclear due to the lack of functional studies and validation in endometriotic tissue, as well as the small sample size of cases harboring *MYH8* mutations. Interestingly, *MYH11*, encoding SM-MHC, was identified as significantly overexpressed in peritoneal endometriosis lesions compared to eutopic endometrium, as reported in multiple studies [59,60]. This upregulation has been associated with tissue remodeling and the differentiation of smooth muscle-like cells. In contrast, *MYH14*, another MYO isoform, was found to be underexpressed in peritoneal lesions, suggesting a shift in cytoskeletal dynamics and contractile properties. h-CALD, a marker for fully differentiated smooth muscle cells, was also confirmed in peritoneal lesions, aiding in distinguishing between smooth muscle hyperplasia and metaplasia.

In adenomyosis, increased levels of SM-MHC, desmin, and oxytocin receptor expression have been observed, reflecting the presence of fully differentiated smooth muscle cells [61]. These markers are strongly associated with fibrosis, suggesting that SM-MHC contributes to ECM remodeling, driven in part by TGF-β released from activated platelets. This highlights the role of smooth muscle differentiation in fibrotic processes, which are shared features of both adenomyosis and endometriosis [61].

Furthermore, MYO6, a unique MYO that moves cargo toward the minus ends of actin filaments, has been implicated in endometriosis pathology. MYO6 was found to be upregulated in endometriotic lesions, suggesting its potential as a biomarker for the disease. Its expression correlated with immune cell infiltration, particularly activated natural killer (NK) cells and M2 macrophages, which are key components of the immune microenvironment in endometriosis [62].

The available literature shows that MYOs play crucial roles in the pathophysiology of endometriosis, contributing to tissue remodeling, fibrosis, and the contractile properties of endometriotic lesions. Their expression closely correlates with the presence of myofibroblasts and smooth muscle-like cells, which are found in different localization of endometriotic lesions. Targeting MYOs-related pathways and smooth muscle contractility represents a promising avenue for therapeutic intervention, particularly for managing fibrosis and pain in endometriosis.

### 2.7. Talin

Talin (TLN) is a key cytoskeletal protein that crosslinks integrins and the actin cytoskeleton, facilitating cell adhesion and signal transduction. It plays a crucial role in forming and stabilizing focal adhesions, which are essential for cell migration and mechanotransduction [63,64].

Tang et al. demonstrated that TLN1 is significantly upregulated in eutopic and ectopic endometrial samples from patients with ovarian endometriotic cysts compared to endometrial tissue from women without endometriosis. The highest levels of TLN1 expression were observed in ectopic endometrium, both at the *TLN1* mRNA and TLN1 protein levels. Notably, the authors cultured stromal cells isolated from ectopic endometrium and found that *TLN1* downregulation via siRNA led to a significant reduction in the cells’ adhesion, migration, and invasion abilities, while proliferation and apoptosis remained unaffected [65].

In contrast, Slater et al. reported a complete absence of TLN expression in endometriotic tissue samples from various localizations. Immunohistochemical analysis showed TLN expression in both the basal and apical regions of uterine epithelial cells in control endometrium from individuals without endometriosis, whereas TLN was undetectable in endometriosis samples. The authors noted minimal variability within each group despite assessing multiple samples (6 controls and 10 endometriosis) [10].

Lee et al. provided mechanistic insights, demonstrating that prostaglandin E2 (PGE_2_) receptors, EP2 and EP4 regulate TLN expression and interactions [66]. Elevated PGE_2_ levels in the peritoneal fluid of women with endometriosis have been previously reported [67]. Selective pharmacological inhibition of EP2 and EP4 reduced TLN expression and disrupted its interactions with other proteins, such as integrins. This, in turn, impaired the adhesion of epithelial and stromal cells derived from endometriotic cysts to extracellular matrix proteins [66].

Ma and Jiang proposed a distinct role for TLN1 in the pathogenesis of endometriosis, highlighting its involvement in the differentiation of regulatory T cells (Tregs) [68]. Disrupted Treg homeostasis, commonly observed in women with endometriosis, is associated with increased systemic and local inflammation in both ectopic and eutopic endometrium [69]. These findings suggest that TLN1 may influence the inflammatory microenvironment of endometriosis by modulating Treg function and differentiation. Mechanistically, TLN1 plays a crucial role in the immunological synapse, where it regulates integrin activation, particularly α4β1 and lymphocyte function-associated antigen 1 (LFA-1), which are essential for Treg adhesion and migration [70]. By stabilizing integrin-mediated signaling cascades, TLN1 influences Treg activation and their ability to suppress inflammation [71]. Furthermore, its interaction with intracellular signaling pathways, such as the Ras-related protein 1 (Rap1)–guanosine triphosphate (GTP)–dependent axis, further regulates T cell activation and differentiation [72]. These mechanisms support the hypothesis that TLN1 contributes to the altered immune response observed in endometriosis.

The current literature reports discrepancies regarding TLN expression in endometriosis, which may reflect variability based on lesion type or localization. Nevertheless, the evidence underscores the critical relationship between inflammatory response, TLN, integrins, and cellular adhesion, highlighting TLN’s multifaceted role in the pathophysiology of endometriosis.

### 2.8. Tensins

Tensins (TNSs) are cytoskeletal proteins that crosslink the actin cytoskeleton and integrin receptors, facilitating cell adhesion, migration, and interaction with the extracellular matrix. They are integral to maintaining cellular architecture and regulating physiological processes such as tissue remodeling and cellular motility. The TNS family consists of four members: tensin-1 (TNS1), tensin-2 (TNS2), tensin-3 (TNS3), and tensin-4 (TNS4), all of which are widely expressed in normal and pathological tissues [73].

TNS1, a cytoplasmic protein located in focal adhesions, plays a critical role in maintaining transmembrane junctions between the cytoskeleton and the extracellular matrix. Cells deficient in TNS1 exhibit significantly reduced migratory capacity, underscoring its role in cell motility [74]. In the female reproductive system, TNS1 has been implicated in processes such as implantation and oogenesis [75]. Moreover, its expression has been detected in the stromal compartment of the endometrium [76].

A study by Rahmawati et al. explored the role of TNS1 in endometriosis, particularly its response to gonadotropin-releasing hormone agonist (GnRHa) therapy, a commonly used treatment for endometriosis [77]. Tissue and serum samples were collected from women with endometriosis who were either untreated or treated with GnRHa. The analysis demonstrated a significant decrease in *TNS1* mRNA and TNS1 protein levels in endometriotic tissue following GnRHa therapy. Immunohistochemistry demonstrated strong TNS1 expression in epithelial and stromal cells of untreated endometriotic tissue, whereas treated samples exhibited markedly reduced expression. Additionally, serum TNS1 concentrations decreased by 53% after GnRHa treatment. These findings suggest that TNS1 may serve as both a therapeutic target and a biomarker for monitoring treatment efficacy in endometriosis [77].

The relevance of TNS1 in endometriosis was further addressed in a commentary by Barra and Ferrero, who highlighted the importance of understanding the molecular mechanisms underlying TNS1’s role in ectopic endometrial cell adhesion and migration [78]. While the study by Rahmawati et al. confirmed the involvement of TNS1 in endometriosis pathology, the authors did not explore its specific contribution to the molecular pathways governing adhesion and migration. Additionally, the commentary noted methodological limitations, such as the exclusive focus on ovarian endometriosis, with no data on other types of lesions like peritoneal or deep infiltrating endometriosis. Furthermore, the specific GnRHa compounds used in the study were not disclosed, despite their varying pharmacokinetic and pharmacodynamic properties [77].

In their response, Rahmawati et al. clarified that their study was limited to patients with ovarian endometriosis and employed a single type of GnRHa. They acknowledged the need for future research to investigate TNS1 expression across different endometriosis phenotypes and to further elucidate its role in disease progression [79].

The evidence presented in the available studies highlights the critical role of TNS1 in cell adhesion and migration, processes that are central to the development and progression of endometriosis. The significant reduction in TNS1 expression following GnRHa therapy underscores its potential as a biomarker for treatment response and a molecular target for therapeutic interventions. Given its role in regulating integrin-mediated signaling and cytoskeletal organization, TNS1 may contribute to the establishment and stabilization of ectopic lesions. Targeting TNS1 or its downstream pathways could impair focal adhesion dynamics, reducing the ability of endometriotic cells to adhere, migrate, and invade surrounding tissues. However, further research is needed to characterize TNS1’s role in other subtypes of endometriosis and to clarify its mechanistic contributions to disease pathophysiology. These findings suggest that TNS1 may offer valuable insights into the pathogenesis and management of endometriosis.

### 2.9. Transgelin

Transgelin (TAGLN) plays a crucial role in regulating cytoskeletal dynamics, cellular contractility, and smooth muscle cell differentiation. It is also associated with processes such as migration and proliferation in various cell types [80].

The available literature indicates that *TAGLN* is significantly upregulated in ectopic endometrium compared to eutopic endometrium of endometriosis patients [81]. A study analyzing 40 matched samples of ectopic and eutopic endometrium from patients with peritoneal or ovarian endometriosis, along with 15 controls without endometriosis, found no difference in *TAGLN* mRNA expression between the eutopic endometrium of endometriosis patients and controls. However, *TAGLN* mRNA expression was significantly higher in ectopic endometrium compared to eutopic endometrium of controls. A limitation of this study was its focus on transcriptome-level analysis, without assessing protein expression [81].

Similarly, another study observed that TAGLN expression was significantly higher in endometriosis lesions compared to normal eutopic endometrium from endometriosis-free controls. However, no statistically significant differences in TAGLN levels were found between paired eutopic and ovarian endometriosis samples [82]. Furthermore, Kyama et al. demonstrated that TAGLN was differentially expressed in endometriotic lesions obtained from peritoneal sites compared to normal peritoneum [83]. Their study identified TAGLN as a specific protein expressed in peritoneal endometriotic lesions but absent in normal peritoneum, highlighting its potential role in the pathogenesis of endometriosis. A common limitation of these studies is their reliance on material isolated from whole tissue samples, without distinguishing between specific cell types. Consequently, it remains unclear whether the observed changes occur in epithelial, stromal, smooth muscle, or endothelial cells, all of which are present in endometriotic lesions.

In contrast, Stephens et al. reported a 4.5-fold decrease in the protein level of transgelin-2 (TAGLN2) in eutopic endometrium of endometriosis patients compared to healthy controls, using MALDI-TOF [84]. However, immunohistochemistry did not show significant differences in the intensity or localization of TAGLN2 between the groups. Immunostaining confirmed the presence of TAGLN2 in all tested samples, with the most intense staining in the luminal and glandular epithelium. Strong staining of the apical surface of cell membranes, with less intense basolateral and cytoplasmic staining, was observed in all patient samples. Minimal TAGLN2 expression was detected in stromal cells, while endothelial cells lining the blood vessels stained intensely, and leukocytes showed a positive signal as well [84].

TAGLN also appears to play a pivotal role in endometriosis pathogenesis as part of the signaling pathway activated by *Fusobacterium nucleatum* infection. This Gram-negative anaerobic bacterium was detected in the uterine tissues of 64% of women with endometriosis, compared to only 7% of controls. *Fusobacterium* infection triggers macrophage infiltration, increased TGF-β production, and subsequent upregulation of transgelin in human and mouse endometria. This leads to enhanced migration, adhesion, and proliferation of endometrial cells, which are crucial processes for the formation and maintenance of endometriotic lesions. In a mouse model of endometriosis, *Fusobacterium* infection promoted lesion development, while antibiotic treatment reduced Tagln expression and significantly decreased lesion number and weight [85,86].

### 2.10. Tropomyosin and Tropomodulin

The tropomyosin (TPM) family consists of proteins that bind along actin filaments in both muscle and non-muscle cells, playing a critical role in cytoskeletal stabilization. In muscle cells, TPM regulates contraction by controlling MYO access to actin binding sites. In non-muscle cells, it stabilizes actin filaments and regulates their interactions with other proteins, impacting essential processes such as cell shape maintenance, motility, and intracellular transport [87].

Tropomodulins (TMODs) are regulatory proteins essential for the stabilization and organization of the actin cytoskeleton. They specifically cap the pointed (minus) ends of actin filaments, preventing further elongation or depolymerization, which contributes to cytoskeletal stability. Together, the TMOD-TPM complex plays a key role in forming and stabilizing dynamic actin structures such as filopodia, lamellipodia, and the cortical actin network, which are critical for cell motility and function [88].

Demir et al. demonstrated that ABPs, including TPM and annexin-1 (ANXA1), may contribute to the ectopic implantation of retrograde menstrual endometrium into the peritoneum. Normally, the peritoneum is lined with mesothelial cells that inhibit ectopic endometrium adhesion as it prefers to interact with ECM components. However, the study showed that menstrual secretion-conditioned medium induced EMT in isolated human omental mesothelial cells, which may lead to cell retraction and exposure of the submesothelial ECM, thus facilitate ectopic implantation. Tropomyosin-α4 (TPM4) and ANXA1 were identified as ABPs with altered phosphorylation patterns in mesothelial cells exposed to the conditioned medium compared to untreated controls [89].

Gajbhiye et al. showed elevated serum levels of antibodies against tropomyosin 3 (TPM3) and tropomodulin 3 (TMOD3) in patients with endometriosis compared to controls [90]. Serological tests using anti-TPM3 and anti-TMOD3 antibodies demonstrated higher sensitivity, specificity, and accuracy in diagnosing early stages of endometriosis (stages I and II) compared to the commonly used marker CA125. The authors highlighted the potential of these antibodies as biomarkers for early detection of minimal-mild endometriosis and that they may be involved in endometriosis-associated infertility [90]. The findings were further supported by a larger study conducted by the same first author yielding consistent results [91]. Similar findings were reported by Menzhinskaya et al., who studied serum from patients with ovarian cysts and deep infiltrative endometriosis and compared it to endometriosis-free controls. Elevated levels of IgM to TPM3 and IgG to TMOD3 were observed in patients with ovarian cysts, while only IgM to TPM3 was elevated in deep infiltrative endometriosis patients. Anti-TPM3 antibodies showed the highest diagnostic value compared to other tested antibodies, with sensitivity and specificity of 73.6% and 81.5%, respectively [92].

However, it is unfortunate that these studies did not investigate the expression and localization of TPM3 and TMOD3 in endometriotic cysts and normal endometrium. It is possible that the observed autoimmune response in endometriosis patients is either a consequence of or a contributing factor to changes in the expression or activity of these proteins in endometrial tissues. This gap in research leaves an important avenue for future studies to explore.

### 2.11. Vinculin

Vinculin (VCL), a focal adhesion protein, connects actin filaments to integrins in the cell membrane, facilitating the transmission of mechanical force and signaling pathways. Wu et al. demonstrated that eutopic and ectopic endometrial stromal cells from patients with endometriosis exhibit higher VCL expression, increased actin polymerization, elevated levels of small GTPases (predominantly ras homolog family member A (RHOA), but also ras-related C3 botulinum toxin substrate 1 (RAC1) and cell division cycle 42 (CDC42)), and enhanced migratory abilities compared to endometrial stromal cells from women without endometriosis. However, several limitations in their analysis should be noted. Fluorescence intensity was not quantified, meaning the conclusions are based solely on visual observations rather than precise measurements. Furthermore, Western blot analyses lacked standard deviation markers and statistical significance annotations, raising concerns about whether the conclusions were drawn from a single measurement or if statistical analysis was omitted entirely. Lastly, the study did not specify whether the groups consisted of single or multiple samples, leaving the robustness of the findings uncertain [93].

Similarly, Börschel et al. observed that a reduced number of VCL plaques correlates with limited migratory abilities of endometrial cells. The authors studied the impact of miRNA-142-3p upregulation on St-T1b cells (an immortalized human endometrial stromal cell line) and endometrial stromal cells isolated from eutopic or ectopic endometrium of various endometriosis localizations. Their results demonstrated that miR-142-3p in St-T1b cells reduced the number and size of VCL-containing focal adhesion plaques and significantly decreased VCL fluorescence intensity compared to controls, alongside a concurrent reduction in cellular migratory abilities. Although *VCL* is not a direct target of miR-142-3p, its downregulation in focal adhesions results from its effects on the cytoskeleton and integrins. miR-142-3p regulates proteins such as Integrin alpha V (ITGAV), Wiskott-Aldrich syndrome like (WASL), RAC1, and Rho GTPases, which are essential for the formation and stabilization of focal adhesion complexes involving VCL [94]. However, interpreting the observed correlation between the reduced number of VCL-containing focal adhesions and diminished cell migratory capacity requires caution. While VCL staining was performed on ST-T1b cells, the migration assay results were based on stromal cells, without clarification on whether these were primary cells or the immortalized cell line. Furthermore, it is worth questioning how well the telomerase-immortalized St-T1b cell line, derived from women without diagnosed endometriosis, models the behavior of stromal cells in eutopic or ectopic endometrium of endometriosis patients [95].

Wu et al. confirmed that in endometrial stromal cells isolated from ovarian cysts, VCL expression decreases in pro-inflammatory conditions stimulated with interleukin-1β (IL-1β). Concurrently, an increase in the migratory ability of the cells was observed. The relationship between VCL expression and the migratory capacity of stromal endometrial cells was further supported by findings in cells treated with both IL-1β and lipoxin A4 (LXA4), an anti-inflammatory mediator. In this group, VCL expression was higher compared to cells treated with IL-1β alone, which corresponded to reduced migratory abilities. These results suggest that VCL may play a role in limiting cell motility by stabilizing adhesion at inflammatory sites, potentially preventing the further spread of endometrial lesions [96]. The study was conducted on samples obtained from six different patients with clinically confirmed endometriosis. After isolation, the researchers verified the correct expression pattern of markers and the proper morphology of endometrial stromal cells. Additionally, changes in expression were validated at both the *VCL* mRNA and VCL protein levels. Unfortunately, the authors of the study did not include endometrial stromal cells from healthy women to compare the effects of the studied compounds and differences in VCL levels.

The available studies show inconsistencies regarding the impact of VCL expression on the migratory abilities of endometriotic stromal cells. However, they collectively indicate that VCL may play a significant role in the development of endometriosis, with external factors such as inflammation and the expression of microRNAs influencing VCL expression in endometrial cells. For this reason, exploring this area further appears to be a promising direction for future research.

## 3. Other ABPs

In this section, we highlight additional ABPs that, while not extensively studied, have been reported to play a role in endometriosis.

### 3.1. Plastins

Plastins (PLS) are a family of ABPs that organize F-actin into complex networks. Among them, T-plastin (PLS3) specifically crosslinks actin filaments into tight bundles and is expressed in various cell types, including epithelial and mesenchymal cells. Its role in actin filament organization makes it vital for cellular processes such as migration, adhesion, and cytoskeletal remodeling. Notably, *PLS3* expression is elevated in the endometrium during the secretory phase in women with minimal or mild endometriosis compared to disease-free controls. Proteomic analyses have identified PLS3 as a potential biomarker, demonstrating 100% sensitivity and 100% specificity for diagnosing minimal to mild endometriosis [97]. These findings suggest PLS3 potential role in the pathogenesis and diagnosis of endometriosis.

### 3.2. Wiskott–Aldrich Syndrome Protein

The Wiskott–Aldrich Syndrome Protein (WASP) family plays a crucial role in actin cytoskeleton remodeling by activating the actin-related protein 2/3 (Arp2/3) complex, a key regulator of cellular processes such as migration and invasion [98]. Among its members, WASP family member 1 (WASF1) has been implicated in the progression of endometriosis. Wakatsuki et al. demonstrated that 17β-estradiol significantly upregulates WASF1 expression, enhancing the migration of immortalized human endometrial stromal cells. In contrast, estetrol (E4), an estrogen receptor modulator, downregulates WASF1 expression and inhibits migration. *WASF1* knockdown further suppressed migration, highlighting its role in facilitating lesion development. These findings propose E4 as a potential therapeutic agent for endometriosis, particularly in combination with progestins, as it may reverse endometriotic changes by modulating WASF1 expression [99].

These observations emphasize the importance of further investigating other ABPs to uncover their functional roles in endometriosis pathophysiology and explore their potential as therapeutic targets or diagnostic biomarkers.

## 4. Angiogenin

Although angiogenin (ANG) is not formally classified as an ABP, actin plays a crucial role in its biological functions. ANG binds to actin in smooth muscle and endothelial cells, forming complexes that activate proteolytic cascades. These cascades degrade laminin and fibronectin in the basement membrane, enabling endothelial cells to penetrate and migrate into perivascular tissues—a key step in angiogenesis [100]. Angiogenesis is essential for the establishment and expansion of endometriotic lesions [101].

Numerous studies have reported elevated ANG levels in the serum and/or peritoneal fluid of endometriosis patients compared to controls [102,103,104]. Singh et al. found significantly higher ANG levels in the serum and follicular fluid of endometriosis patients [102], while Bourlev et al. observed elevated levels in serum and peritoneal fluid in advanced endometriosis cases, with no significant reduction post-surgical resection [103]. Steff et al. reported that ANG levels were higher during the follicular phase but showed no differences in the luteal phase. ANG levels correlated positively with disease severity and the number of poorly vascularized (white) lesions, but negatively with necrotic (black) lesions, showing no correlation with hypervascularized (red) lesions. These findings suggest ANG’s role in vascularizing white lesions [104]. Similarly, Suzumori et al. reported higher peritoneal fluid ANG levels in endometriosis patients, with no differences across menstrual phases or between patients with and without red lesions [105].

In contrast, Gogacz et al. found that ANG levels in peritoneal fluid and blood were not reliable markers for endometriosis. Controls with idiopathic infertility showed higher ANG levels in peritoneal fluid compared to endometriosis patients, with no significant differences in blood levels [106].

At the cellular level, Kim et al. observed that *ANG* mRNA expression was downregulated in ectopic endometrium of endometriosis patients during the mid-secretory phase. In healthy controls, *ANG* expression increased between proliferative and secretory phases, aligning with the implantation window, whereas this pattern was absent in endometriosis patients [107]. Fu et al. demonstrated higher expression in stromal cells from ectopic endometrium compared to eutopic endometrium at both *ANG* mRNA and ANG protein levels. Ectopic stromal cells also secreted more ANG, which stimulated human umbilical vein endothelial cells (HUVECs) to form more tubular structures. Immunohistochemical analysis showed colocalization of ANG-positive regions with CD31-positive endothelial cells in ectopic endometrium, indicating its role in vascularization of endometriotic tissue [108].

Further evidence of angiogenesis-stimulating conditions in ectopic endometrium comes from studies on ANG’s inhibitor, Ribonuclease/Angiogenin Inhibitor 1 (RNH1). Immunohistochemical analysis identified significantly lower RNH1 levels in eutopic glandular epithelium of endometriosis patients compared to controls, indicating reduced inhibition of angiogenesis. However, protein quantification via Western blot was inconclusive [84].

In summary, endometriosis patients often exhibit elevated ANG levels in serum and peritoneal fluid, likely driven by hypoxia and angiogenic stimulation, particularly in poorly vascularized white lesions. Additionally, the lack of cyclical changes in ANG expression between proliferative and secretory phases in eutopic endometrium may impair embryo implantation and contribute to infertility. These findings highlight ANG’s critical role in endometriosis pathogenesis and associated symptoms.

## 5. Summary

ABPs are essential regulators of cellular homeostasis. Through their interactions with the actin cytoskeleton, ABPs play critical roles in processes such as cell division, migration, and apoptosis. Alterations in ABPs expression, activity, or intracellular localization can lead to actin cytoskeleton reorganization, which ultimately governs cellular behavior. Given these roles, the involvement of ABPs in the progression of endometriosis is highly plausible, as highlighted by available research.

Existing studies suggest several key mechanisms through which ABPs contribute to the development and progression of endometriosis. These include enhancing cellular migration (ACTN1, ERM, *FSCN1*) and adhesion (TLN, TNS, VCL), promoting fibroblast-to-myofibroblast transition (CNN, MYO, CALD, TAGLN), and facilitating the formation of vascular networks essential for the growth and survival of ectopic endometrial lesions (ANG). Given the criticality of these processes to the pathophysiology of endometriosis, ABPs represent promising therapeutic targets for managing the disease and its associated symptoms, such as pain and infertility (Figure 2, Table 1).

However, significant challenges arise when analyzing the existing literature due to methodological inconsistencies and a lack of systematic approach. The diversity of study designs and experimental models leads to variability in findings. For example, many studies rely on tissue samples from patients with endometriosis, often comparing ectopic endometrium (from lesions) to eutopic endometrium (from the same patient or disease-free controls). Some studies use matched samples, while others include control tissues from women with unrelated gynecological conditions. In other cases, immortalized cell lines, such as the 12Z line derived from endometrial glands, are used. While cell lines offer practical advantages for preliminary research, they fail to capture the heterogeneity of endometriotic lesions and the critical interactions between cells and their microenvironment.

Another limitation lies in the variability of control groups and the lack of standardization in experimental methodologies. Some studies analyze ABP expression in ectopic endometrial lesions relative to surrounding tissues, such as the peritoneum, rather than normal endometrium. In studies using primary cell cultures, stromal cells from endometrial lesions are often the focus, but the validation of stromal cell identity varies. While some studies rigorously confirm cell markers, others rely solely on morphological assessments. Furthermore, significant variability exists in the aspects of ABP biology investigated—some studies assess mRNA expression, others focus on protein levels, post-translational modifications (e.g., phosphorylation), or intracellular localization. Without a comprehensive evaluation of these factors, drawing conclusions about the role of specific ABPs in endometriosis progression remains challenging.

These methodological inconsistencies have led to conflicting findings. For example, TLN has been reported to exhibit both significant upregulation in endometriotic lesions compared to normal endometrium [65] and complete loss of expression in other studies [10]. Such discrepancies highlight the need for systematic approaches to elucidate the role of ABPs in endometriosis.

Additionally, research demonstrates variability in ABP behavior depending on the localization of endometriotic lesions. For instance, ovarian, peritoneal, and other ectopic sites may exhibit distinct molecular profiles. Genomic differences between cells from endometriosis and normal endometrium further complicate the analysis. While some studies explore ABPs as therapeutic targets, others focus on their potential as diagnostic markers, often analyzing ABP or anti-ABPs antibodies levels in peripheral blood or peritoneal fluid. Such diversity in study objectives, materials, and methods underscores the need for more standardized research approaches.

## 6. Future Perspective

After conducting a thorough literature review, we identified five key recommendations to guide future studies. To enhance our understanding of the role of ABPs in endometriosis, it is essential for future research to adopt more systematic and comprehensive approaches. Given the crucial role of inflammation in the pathophysiology of endometriosis, further studies should explore how ABPs interact with inflammatory triggers, including cytokines, growth factors, and immune cells, which are known to affect cytoskeletal remodeling [109,110]. Our recommendations are as follows:Standardized Sample Selection: Control and experimental groups should be well-defined and matched based on parameters such as age, menstrual cycle phase, and hormonal treatment history. Ideally, studies should include three types of samples: ectopic endometrium, eutopic endometrium from women with endometriosis, and eutopic endometrium from disease-free individuals. This approach could help elucidate differences in endometrium between women with and without endometriosis and provide insights into the mechanisms initiating the disease.Comprehensive Analysis of ABP Function: Studies should evaluate ABP expression at multiple levels, including mRNA, protein, post-translational modifications, and intracellular localization. Integrating these data will provide a more holistic understanding of ABP involvement in endometriosis progression.Interplay Between ABPs and Inflammatory Processes: Inflammation is a central factor driving the pathogenesis of endometriosis, and inflammatory mediators such as IL-1β, TNF-α, and TGF-β are known to influence cytoskeletal remodeling. Future studies should investigate how these cytokines regulate ABP function and whether changes in ABP expression contribute to immune evasion, lesion stability, or fibrosis in endometriosis. Furthermore, co-culture models incorporating immune cells or inflammatory mediators should be considered to better mimic the in vivo microenvironment of endometriotic lesions.Balanced Use of Models: While immortalized cell lines, such as 12Z, are valuable for preliminary, high-throughput studies, their limitations (lack of heterogeneity and absence of microenvironmental interactions) must be acknowledged. Findings from cell lines should be further validated using primary cells or tissue samples to ensure biological relevance.Validation of Primary Cell Cultures: In studies using primary cell cultures, the identity of isolated cells should be rigorously validated using characteristic markers. This ensures consistency and reliability in interpreting results.Exploration of Diagnostic and Therapeutic Potential: Future research should investigate the dual roles of ABPs as both diagnostic markers and therapeutic targets. For diagnostic applications, blood and peritoneal fluid may offer non-invasive sources for identifying ABP-based biomarkers. For therapeutic purposes, targeting specific ABPs involved in key processes, such as migration, adhesion, or angiogenesis, could yield novel strategies for managing endometriosis and its symptoms.

By addressing these gaps, future research has the potential to uncover the precise mechanisms by which ABPs contribute to the progression of endometriosis. We strongly believe that this knowledge could pave the way for novel therapeutic approaches, improved diagnostic tools, and a deeper understanding of the disease’s pathophysiology.

## Figures and Tables

**Figure 1 cells-14-00360-f001:**
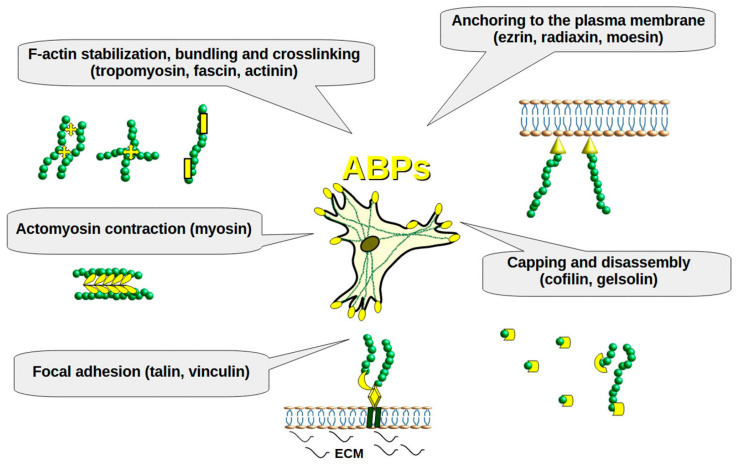
The roles of ABPs in the cell. F-actin- green, ABPs-yellow.

**Figure 2 cells-14-00360-f002:**
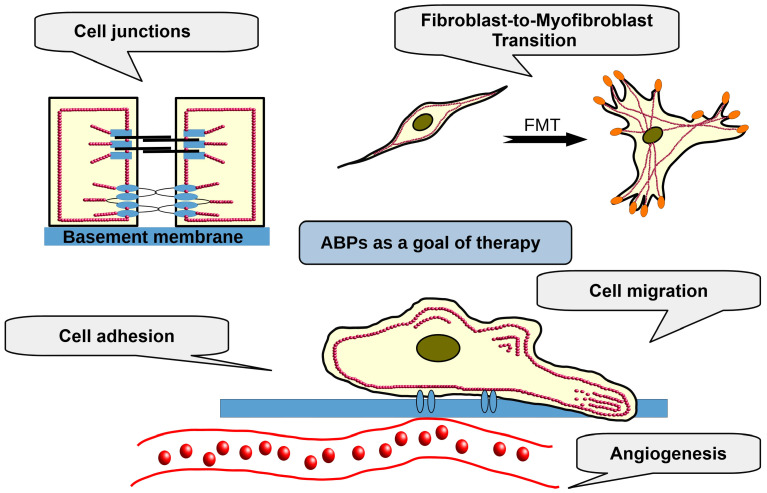
ABPs as a goal of therapy.

**Table 1 cells-14-00360-t001:** Summary of the involvement of ABP proteins in endometriosis. WB—Western blot, IHC—immunohistochemistry, PCR—polymerase chain reaction.

Protein	Control Group	Study Group	Conclusions	Comments	Reference
Alpha-actinin (ACTN)	Eutopic endometrium from endometriosis patient (*n* = 1)	Ectopic endometrium from endometriosis patient (*n* = 1)	ACTN1 is significantly upregulated in ectopic endometrium. There is no significant difference in ACTN4 level between samples.	This is a single patient study. ACTN1 was studied at mRNA (PCR) and protein level (WB).	[14]
	Eutopic endometrium from endometriosis-free controls (*n* = 6)	Ectopic endometrium from different localizations (*n* = 15)	ACTN1 is significantly upregulated in ectopic endometrium, with its expression levels increasing proportionally to the distance from eutopic endometrial tissue. There are no significant differences in ACTN4 levels between groups.	ACTN1 was investigated at both the mRNA (PCR) and protein levels (WB). The study also encompassed cases of rare endometriosis localizations, including umbilical endometriosis.	[15]
	Eutopic endometrium from endometriosis-free controls in proliferative or midsecretory phases (*n* = 6)	Ectopic endometrium from different localizations (*n* = 10)	ACTN is deexpressed in ectopic endometrium. In eutopic endometrium its expression occurs in the apical part of the glandular epithelium.	The isoform of ACTN was not specified in the study. ACTN expression was assessed exclusively at the protein level (IHC). The article does not provide sufficient characterization of the study and control groups.	[10]
Calponin (CNN)	Eutopic endometrium from endometriosis patients (*n* = 7)	Ectopic endometrium from ovarian lesions (*n* = 6)	CNN1 is significantly upregulated in ectopic endometrium.	The study includes extensive proteomic analysis. Positive CNN1 staining was confirmed in most, but not all ectopic endometrium samples.	[23]
	Peritoneum from endometriosis-free controls (*n* = 10) or endometriosis patients but distant from endometriotic lesion (*n* = 5)	Ectopic endometrium from peritoneal lesions (*n* = 23)	CNN is upregulated in endometriosis ectopic endometrium. Its level is the highest in the central compartment of the lesion.	CNN was studied only at the protein level (IHC). The samples were collected in different phases of menstrual cycle, which did not affect the CNN expression in endometriosis lesions.	[24]
Cofilin (CFL)	Eutopic endometrium from endometriosis-free baboons (*Papio anubis*) in proliferative and secretory phase (*n* = 6)	Eutopic endometrium from baboons (*Papio anubis*) with induced endometriosis in proliferative secretory phase (*n* = 3)	During the secretory phase, CFL changes cellular localization in endometrium of controls but not in study group. It may explain the difficulties in the embryo implantation and high infertility rates connected with endometriosis.	The results have yet to be validated through studies on human tissues. CFL was analyzed at both the mRNA (PCR) and the protein level (IHC).	[11]
	Stromal cells isolated from eutopic endometrium of endometriosis-free controls	Stromal cells isolated from ovarian endometriosis lesions (*n* = 30)	CFL1 is upregulated in stromal cells from ectopic endometrium. CFL1 silencing in ectopic endometrial stromal cells limits cell proliferation, adhesion, invasion, angiogenesis, and stimulates apoptosis.	All tissue samples were collected in secretory phase confirmed thorough histological evaluation. The purity of primary cultures was confirmed using markers (vimentin, CD10, cytokeratin, factor VIII and leukocyte common antigen).	[25]
	Eutopic endometrium of endometriosis-free controls in secretory phase (*n* = 10)	Eutopic endometrium of endometriosis patients in secretory phase (*n* = 10)	*CFL1* is upregulated in eutopic endometrium of endometriosis patients.	*CFL1* was analyzed at the mRNA level. The secretory phase was confirmed through histological evaluation.	[26]
	Stromal cells isolated from eutopic endometrium of endometriosis patients in secretory phase (*n* = 20)	Stromal cells isolated from eutopic endometrium of endometriosis patients in secretory phase treated with estradiol (*n* = 20)	Estradiol mediates the phosphorylation of CFL1 via LIM domain kinase 1.	All stromal tissue samples were collected in secretory phase confirmed thorough histological evaluation. The purity of primary cultures was confirmed using markers (vimentin, CD10, cytokeratin, factor VIII and leukocyte common antigen).	[27]
	Stromal cells isolated from eutopic endometrium of endometriosis patients in secretory phase (*n* = 15)	Stromal cells isolated from eutopic endometrium of endometriosis patients in secretory phase after cofilin-1 silencing and/or platelet-derived growth factor treatment (*n* = 15)	Silencing of CFL1 significantly reduces proliferative effect of platelet-derived growth factor.	All stromal tissue samples were collected in secretory phase confirmed thorough histological evaluation. The purity of primary cultures was confirmed using markers (vimentin, CD10, cytokeratin, factor VIII and leukocyte common antigen).	[29]
	12Z cell line (endometriotic epithelial cells)	12Z cell line after long intergenic noncoding RNA 01,133 knockdown	Knockdown of long intergenic noncoding RNA 01,133 causes increased CFL phosphorylation.	The isoform of CFL was not specified. Phosphorylation of CFL at Ser-3 is crucial regulatory mechanism.	[32]
Ezrin (EZR) Radixin (RDX) Moesin (MSN) (ERM)	Eutopic endometrium of endometriosis-free controls in proliferative or midsecretory phases (*n* = 6)	Ectopic endometrium from different localizations (*n* = 10)	Endometriotic lesions are characterized by a complete loss of EZR expression.	EZR expression was assessed exclusively at the protein level (IHC). The article does not provide sufficient characterization of the study and control groups.	[10]
	Eutopic endometrium of endometriosis-free controls in proliferative phase (*n* = 12)	Ectopic and eutopic endometrial tissues from endometriosis patients with lesions in various localizations during the proliferative phase (*n* = 13)	A progressive increase in EZR and pEZR levels occurs from controls to eutopic and ectopic samples, correlating with enhanced cellular invasiveness.	In tissue samples, EZR was analyzed exclusively at the protein level (IHC). Further studies involved the isolation of stromal cells, which were subsequently used as material for Western blot analysis and invasion assays.	[33]
	Eutopic endometrium of endometriosis-free controls in proliferative phase (*n* = 30)	Ectopic and eutopic endometrial tissues from endometriosis patients with ovarian lesions during the proliferative phase (*n* = 49, *n* = 32, respectively)	EZR expression increases progressively from controls to eutopic and ectopic endometrium and is present in both glandular and stromal cells. Inhibiting EZR phosphorylation in stromal cells from ectopic endometrium reduced pEZR levels, impaired migratory structures, and decreased cellular migration and invasion.	The study involved both tissue samples and the isolation of endometriotic stromal cells. EZR phosphorylation, essential for its activity, was inhibited in the study using NSC305787.	[34]
	-	Ectopic endometrium from women with endometriosis across various localizations and menstrual cycle phases (*n* = 57)	There are no differences in EZR or pEZR expression and localization between endometriosis tissue samples from different sites or phases of the menstrual cycle.	The study did not include any control samples. Differences were assessed between various types of endometriotic lesions collected during different phases of the menstrual cycle. EZR and pEZR were analyzed exclusively at the protein level (IHC).	[35]
	Eutopic endometrium of endometriosis-free controls in proliferative or secretory phase (*n* = 15)	Ectopic and eutopic endometrium from women with endometriosis in proliferative or secretory phase (*n* = 14)	*EZR* expression gradually increases from control to eutopic and ectopic endometrium, regardless of menstrual cycle phase. *EZR* downregulation in stromal cells from ectopic endometrium significantly reduces migratory potential.	*EZR* levels in tissue samples were assessed exclusively at the mRNA level, while the stromal nature of the isolated cells was confirmed through vimentin labeling.	[36]
	Eutopic endometrium of endometriosis-free controls in proliferative phase (*n* = 10)	Ectopic endometrium from different localizations (*n* = 23) 12Z cell line (endometriotic epithelial cells)	Sphingosine-1-phosphate receptor 3 expression is upregulated in endometriosis lesions, which correlates with EMT and fibrosis markers. Its effects are abolished by the EZR inhibitor NSC668394.	EZR and sphingosine-1-phosphate receptor 3 are functionally interconnected through their roles in cell signaling, cytoskeletal organization, and cellular migration, particularly in processes like cancer metastasis, inflammation, and tissue remodeling. Studies were partially conducted on 12Z cell line and are yet to be evaluated in primary cultures.	[37]
	Stromal cells isolated from eutopic endometrium of endometriosis-free controls in proliferative phase (*n* = 14)	Stromal cells isolated from the eutopic and ectopic endometrium of ovarian lesions in endometriosis patients (*n* = 16, *n* = 8, respectively)	Elevated ERM protein phosphorylation in stromal cells from ectopic endometrium, regulated by the Raf-1/ROCKII pathway, is associated with contracted morphology, reduced migratory potential, and stabilization of lesions at implantation sites.	The purity of the stromal cell culture was evaluated by immunofluorescence analysis using antibodies against vimentin, cytokeratin7, and CD45.	[38]
	Stromal cells isolated from eutopic endometrium from endometriosis-free controls (*n* = 2)	Stromal cells isolated from the eutopic and ectopic endometrium from endometriosis patients (*n* = 4)	Sorafenib treatment reduces the pEZR-to-EZR ratio, correlating with decreased migratory but not invasive properties.	Isolated stromal cells were well-characterized using multiple cellular markers.	[39]
	hEM15A (immortalized stromal cell line resembling eutopic stromal cells from women with endometriosis)	hEM15A with extracellular matrix protein 1 downregulation	ECM1 downregulation decreases RDX expression via the RhoC/ROCK1 pathway.	The study also included tissue samples; however, RDX was analyzed using an in vitro cellular model. These observations have yet to be confirmed in primary cell cultures.	[40]
	Endometrial fluid from endometriosis-free controls (*n* = 32)	Endometrial fluid from endometriosis patients (*n* = 46)	Endometrial fluid provides a valuable and non-invasive sample for proteomic analysis, enabling the identification of disease-specific proteins associated with endometriosis.	This study expands the list of potential diagnostic markers by identifying 31 differentially expressed proteins, including MSN.	[41]
	Stromal cells from eutopic endometrium and vaginal secretion from endometriosis-free controls (*n* = 10); Ovarian epithelium isolated form endometriosis-free controls (*n*—not given); Human umbilical vein endothelial cells (HUVECs)	Stromal cells from ovarian endometriotic lesions and vaginal fluid from endometriosis patients (*n* = 6)	Exosomes from endometriotic stromal cells, mediated by MSN, promote normal stromal cell migration, endothelial tubular structure formation, and inflammatory cytokine production in ovarian epithelial cells, significantly affecting surrounding normal cells.	This study provides valuable insights into the impact of extracellular vesicles on the behavior of various cell types involved in endometriosis development. The vesicles were well-characterized, and the use of diverse methodologies enhances the reliability of the findings.	[42]
Fascin (FSCN)	Eutopic endometrium of endometriosis-free controls (*n*—not given); CRL-7566 cell line (endometriotic cell line)	Ectopic endometrium from ovarian lesions (*n* = 20); CRL-7566 cells with autophagy protein 5 downregulation	*FSCN1* is upregulated in ectopic endometrium, playing a key role in filopodia formation and enhanced invasiveness, as its exogenous upregulation reverses the suppressive effects of autophagy activation on proliferation, migration, and invasiveness.	The conclusions were partially based on data from a cell line model; further validation in primary cell cultures is necessary to strengthen the findings.	[49]
	Stromal cells isolated from eutopic (*n* = 3) or ectopic (*n* = 4) endometrium; 12Z cell line (endometriotic epithelial cells)	Stromal cells isolated from eutopic (*n* = 3) or ectopic (*n* = 4) endometrium after miR-145 upregulation. 12Z cell line after miR-145 upregulation.	Overexpression of miR-145 downregulates *FSCN1* leading to decreased invasiveness and motility in both the 12Z endometriotic cell line and primary stromal cells.	The studies were partially performed on 12Z cell line and need to be further validated in more heterogeneous primary cultures.	[50]
Myosin (MYO) and caldesmon (CALD)	Peritoneum of endometriosis-free controls (*n* = 10)	Peritoneum and ectopic endometrium of endometriosis patients in different menstrual cycle phases (*n* = 60)	Smooth muscle-like cells in endometriosis express contractile markers and receptors for vasopressin and oxytocin, suggesting their ability to contract in response to hormonal stimulation, potentially triggering nociceptors and contributing to pelvic pain in endometriosis.	The study focuses more on understanding the development of endometriosis-associated symptoms rather than the mechanisms underlying the disease itself. This approach could provide a foundation for developing symptomatic treatment strategies.	[53]
	Eutopic endometrium of endometriosis-free controls (*n* = 50)	Eutopic and ectopic endometrium from ovarian lesions of endometriosis patients (*n* = 50)	Ovarian endometriotic lesions exhibit strong positive expression SM-MHC, desmin, and α-smooth muscle MYO.	Although the study does not focus specifically on the role of MYO in endometriosis, it supports observations made by other researchers. It incorporates a broader range of materials, including cell lines and an in vivo mouse model, primarily to investigate Serine/threonine-protein kinase Pim-2 (Proviral Integrations of Moloney virus 2).	[56]
	Eutopic endometrium of endometriosis-free controls (*n* = 2)	Ectopic endometrium from deep infiltrating endometriosis lesions (*n* = 20); Nude mouse model with induced endometriosis (*n* = 8)	Deeply infiltrating endometriosis lesions are characterized by fibromuscular tissue with myofibroblasts expressing smooth muscle markers such as α- smooth muscle MYO, desmin, and SM-MHC. The study suggests that these cells arise from the transdifferentiation of local fibroblasts in response to ectopic endometrium.	An estradiol release capsule was used to regulate the cycle of the mice, maintaining stable physiological estrogen levels and eliminating interspecies differences with the human menstrual cycle. Implantation of ectopic endometrium did not occur in one of the eight mice, reducing the final sample size of the study group to seven.	[57]
	Genomic DNA extracted from granulocytes of Han Chinese endometriosis-free controls (*n* = 485)	Genomic DNA extracted from granulocytes of Han Chinese women diagnosed with ovarian endometriosis (*n* = 152)	The study identified two novel *MYH8* mutations in patients with ovarian endometriosis, suggesting a potential role of these mutations in the pathogenesis of endometriosis through mechanisms involving cell migration and invasion.	The use of granulocyte-derived genomic DNA provided a non-invasive approach for genetic analysis, but the lack of functional studies and validation in endometriotic tissue remains a limitation of the study.	[58]
	Gene expression data from normal endometrial tissues from GEO datasets GSE25628, GSE5108, and GSE7305	Gene expression data from endometriotic tissues in GEO datasets GSE25628, GSE5108, and GSE7305	A total of 119 differentially expressed genes and 5 hub genes were identified, including smooth muscle *MYO*.	The study used a robust bioinformatics approach but lacked experimental validation of the identified genes and pathways, limiting direct clinical applicability.	[60]
	Eutopic endometrium and peritoneum from patients with peritoneal endometriosis (*n* = 17 and *n* = 22, respectively)	Ectopic endometrium from peritoneal lesions (*n* = 18)	Significant differences in gene expression related to cytoskeletal remodeling and smooth muscle contraction were observed. *MYH11* was one of the most significantly overexpressed genes in peritoneal lesions, while *MYH14* was one of the most suppressed. CALD acts as a key marker to distinguish hyperplasia from metaplasia.	The study highlights distinct gene expression patterns and tissue remodeling in peritoneal endometriosis but lacks functional studies to validate the identified pathways and molecular mechanisms.	[59]
	Eutopic endometrium of endometriosis-free controls (*n* = 20)	Eutopic and ectopic endometrium from adenomyotic lesions (*n* = 34)	The study highlights the involvement of SM-MHC in adenomyotic lesions, supporting its role in fibrogenesis and tissue remodeling.	The study employs various staining techniques and experiments.	[61]
	Gene expression data from normal endometrial tissues in GEO databases (GSE7305 and GSE1169)	Gene expression data from endometriosis patients in GEO databases (GSE7305 and GSE1169)	*MYO6* expression is increased in endometriosis and correlates with immune cell infiltration, suggesting its role in endometriosis pathology.	The study is limited to bioinformatic analysis and lacks functional validation of *MYO6* in experimental models.	[62]
Talin (TLN)	Eutopic endometrium of endometriosis-free controls in proliferative phase (*n* = 15)	Eutopic and ovarian ectopic endometrium samples from endometriosis patients in the proliferative phase (*n* = 26)	TLN1 is significantly upregulated in endometriosis, while its downregulation in endometriotic stromal cells reduces cellular adhesion, migration, and invasion.	TLN1 was assessed at mRNA (PCR) and protein (IHC) levels. TLN1 downregulation had no effect on the proliferation or cell death of endometriotic stromal cells.	[65]
	Eutopic endometrium of endometriosis-free controls in proliferative or midsecretory phases (*n* = 6)	Ectopic endometrium from different localizations (*n* = 10)	TLN expression was absent in endometriotic tissues from various localizations, while control endometrium showed consistent TLN expression in both basal and apical regions of uterine epithelial cells.	The isoform of TLN was not specified in the study. TLN expression was assessed exclusively at the protein level (IHC). The article does not provide sufficient characterization of the study and control groups.	[10]
	12Z (endometriotic epithelial cells) and 22B (endometriotic stromal cells) cell lines	12Z and 22B cells treated with specific E2 or E4 inhibitors	Inhibition of E2 and E4 resulted in decreased TLN expression, disrupted interactions with integrins, and impaired cell adhesion to ECM components.	The study utilizes well-characterized endometriotic cell lines, validated for their ability to form endometriosis lesions in mice; however, the findings still require confirmation using more heterogeneous primary cell lines derived from patient samples.	[66]
	Peritoneal fluid from endometriosis-free controls (*n* = 20)	Peritoneal fluid from endometriosis patients (*n* = 20)	TLN1 plays a role in the differentiation of regulatory T cells, whose dysregulation is commonly observed in women with endometriosis.	The authors carried out extensive experiments utilizing various sources of material, including tissue samples; however, TLN1 expression was solely analyzed in monocytes derived from peritoneal fluid.	[68]
Tensin (TNS)	Ectopic endometrium and serum samples from ovarian endometriosis patients (*n* = 30)	Ectopic endometrium and serum samples of ovarian endometriosis patients treated with gonadotropin-releasing hormone agonist (*n* = 29)	Gonadotropin-releasing hormone agonist therapy significantly reduces TNS1 expression in endometriotic tissues and serum, highlighting its potential as a therapeutic target and biomarker for monitoring treatment efficacy in endometriosis.	TNS1 was thoroughly investigated using multiple methods at both the mRNA (PCR) and protein levels (IHC, WB, ELISA). The study sparked a discussion regarding the clinical relevance of the findings, prompting the authors to acknowledge the need to expand the research to include patients with lesions in other locations.	[77,78,79]
Transgelin (TAGLN)	Eutopic endometrium from endometriosis-free controls (*n* = 15; two biopsies per patient—one in the proliferative and one in the secretory phase).	Matched samples of eutopic and ectopic endometrium from endometriosis patients in the proliferative or secretory phase, with varying lesion localizations. (*n* = 40)	*TAGLN* mRNA expression does not differ between the eutopic endometrium of endometriosis patients and controls but is significantly upregulated in ectopic endometrium, suggesting its potential involvement in lesion formation.	*TAGLN* was analyzed only at the mRNA level (PCR). Further studies are necessary to validate these findings at the protein level and explore the functional role of TAGLN in endometriotic tissue.	[81]
	Eutopic endometrium from endometriosis-free control (*n* = 1)	Matched samples of eutopic and ectopic endometrium from endometriosis patients (*n* = 8)	TAGLN expression is significantly elevated in endometriotic lesions compared to eutopic endometrium from endometriosis-free controls, but no significant differences were observed between paired eutopic and ovarian endometriosis samples.	The study incorporated a comprehensive proteomic analysis; however, the conclusions are limited by the small sample size in both the control and study groups.	[82]
	Eutopic endometrium from endometriosis-free controls (*n* = 3)	Eutopic endometrium, ectopic endometrium from peritoneal lesions, and macroscopically normal peritoneum from endometriosis patients (*n* = 3).	TAGLN is a specific protein present in peritoneal endometriotic lesions but absent in normal peritoneum, suggesting its potential involvement in the pathogenesis of endometriosis.	Further studies are needed to determine the cellular origin of TAGLN expression within endometriotic lesions, as current findings are based on whole tissue analysis without distinguishing between epithelial, stromal, smooth muscle, or endothelial cells.	[83]
	Eutopic endometrium from endometriosis-free controls (*n* = 4)	Eutopic endometrium from endometriosis patients (*n* = 4)	MALDI-TOF analysis indicated a significant reduction in TAGLN2 protein levels in the eutopic endometrium of patients with endometriosis compared to healthy controls, but immunohistochemistry did not confirm these differences in staining intensity or localization.	The study highlights the need for complementary validation techniques, as discrepancies between proteomic quantification and immunohistochemistry raise questions about the functional relevance of TAGLN 2 in endometriosis and its specific cellular distribution.	[84]
Tropomyosin (TPM) and tropomodulin (TMOD)	Human omental mesothelial cells	Human omental mesothelial cells cultured in conditioned medium derived from menstrual effluent	The altered phosphorylation patterns of TPM4 and ANXA1in mesothelial cells exposed to conditioned medium highlight a possible mechanism by which cytoskeletal remodeling contributes to endometriosis pathogenesis.	The study incorporates a comprehensive proteomic analysis utilizing multiple methods.	[89]
	Serum from endometriosis-free controls (*n* = 30) Eutopic endometrium from endometriosis-free controls in secretory phase (*n* = 27)	Serum from endometriosis patients (*n* = 40) Eutopic endometrium from endometriosis patients in secretory phase (*n* = 18)	Elevated serum levels of anti-TPM3 and anti-TMOD3 antibodies in endometriosis patients suggest their potential as more sensitive and specific biomarkers for early-stage diagnosis, outperforming CA125 and potentially linking them to endometriosis-associated infertility.	Tissue expression and localization of TPM3 and TMOD3 were not studied.	[90]
	Serum from endometriosis-free controls (*n* = 104)	Serum from endometriosis patients (*n* = 133)	The concurrent measurement of antibodies against TMOD3 and TPM3 isoforms may be used as a noninvasive biomarker panel for diagnosing minimal–mild endometriosis.	The authors admit that further validation is needed to assess its clinical utility in detecting endometriosis before laparoscopy and monitoring disease progression and treatment response.	[91]
	Serum from endometriosis-free controls (*n* = 27)	Serum from endometriosis patients (*n* = 74)	Elevated IgM to TPM3 and IgG to TMOD3 in serum of patients with ovarian cysts, along with increased IgM to TPM3 in deep infiltrative endometriosis, suggest their potential role as biomarkers. Anti-TPM3 antibodies demonstrate the highest diagnostic value for endometriosis detection.	As the study focuses on potential endometriosis biomarkers and does not include tissue samples, the expression and localization of TPM3 and TMOD3 in tissues were not inve109stigated.	[92]
Vinculin (VCL)	Endometrial stromal cells from endometriosis-free controls	Endometrial stromal cells isolated from eutopic and ectopic endometrium	Increased VCL expression in stromal cells from endometriosis patients is associated with enhanced migratory capacity, suggesting its potential role in disease progression.	This study has several limitations, including the lack of fluorescence intensity measurements, issues with statistical analysis and data presentation, and the absence of information regarding sample size.	[93]
	St-T1b cell line (telomerase-immortalized human endometrial stromal cells), endometrial stromal cells isolated from eutopic and ectopic endometrium of endometriosis patients	St-T1b cell line with miR-142-3p upregulation, endometrial stromal cells isolated from eutopic and ectopic endometrium of endometriosis patients with miR-142-3p upregulation	miR-142-3p reduces the number and size of VCL-containing focal adhesions, significantly decreases VCL fluorescence intensity, and impairs migratory abilities in St-T1b cells, suggesting its potential role in regulating cell adhesion and motility.	Studies were partially conducted on the immortalized St-T1b cell line. There is no information on whether the stromal nature of the cells was confirmed using specific marker labeling.	[95]
	Endometrial stromal cells isolated from ovarian endometriosis lesions (*n* = 6)	Endometrial stromal cells isolated from ovarian endometriosis lesions treated with interleukin-1β or interleukin-1β and lipoxin A4 or lipoxin A4 (*n* = 6)	VCL may play a role in limiting cell motility by stabilizing adhesion at inflammatory sites.	The study includes validation of endometrial stromal cell markers and assesses VCL expression at both the mRNA and protein levels.	[96]
Plastin (PLS)	Eutopic endometrium from endometriosis-free controls in secretory phase (*n* = 10)	Eutopic endometrium of endometriosis patients in secretory phase (*n* = 19)	PLS3 is upregulated in the secretory phase endometrium of women with minimal to mild endometriosis and has been identified as a potential biomarker with 100% sensitivity and specificity for disease detection.	Further validation in larger, independent cohorts is necessary to confirm its clinical utility and investigate its functional role in disease pathogenesis	[97]
Wiskott–Aldrich Syndrome Protein (WASP)	Immortalized human endometrial stromal cells	Immortalized human endometrial stromal cells treated with 17β estradiol or estetrol	WASP family member 1 plays a key role in endometrial stromal cell migration, as its expression is upregulated by 17β-estradiol, promoting increased motility, while estetrol downregulates WASP family member 1 and inhibits migration.	Since this study was conducted on an immortalized human endometrial stromal cell line, the findings require validation in more heterogeneous primary cell lines to confirm the role of WASP family member 1 in endometriosis pathogenesis and its potential as a therapeutic target.	[99]

## Data Availability

Not applicable.

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
