# Peer review of "Endometriosis and Cytoskeletal Remodeling: The Functional Role of Actin-Binding Proteins"

_cells, 2025, doi:10.3390/cells14050360_

Round 1
Reviewer 1 Report
Comments and Suggestions for Authors
In the present review, Arendt et al provide an excellent comprehensive review of the putative role of actin binding proteins in the pathogenesis of endometriosis and potential relevance as therapeutic targets or mechanisms which may be amenable to manipulation to improve endometriosis outcomes indirectly.
The prose of the review is very well written, well-referenced, and up to date for many members/families of the ABP interactome which will be of interest to anyone who studies endometriosis.
The introduction well summarizes the problems and prevalence of endometriosis which is a horrible debilitating disease. I often see endometriosis papers start off with it’s a benign disease (since it’s not cancer), but that couldn’t be more wrong, I assure you it’s not benign to those who suffer from it. The numbers of women who likely have lesions that alter their fertility or quality of life is probably underestimated because of the range of symptoms, timing, and reliance of laparoscopy to properly diagnose which only a subset seek out. Thus, it’s a timely review that speak to a relevant health concern.
I don’t have any major issues with the organization or quality of the review (i.e., literature cited, depth of studies examined, etc.).
Some minor points to consider.
Protein/gene nomenclature floats a bit throughout the document (If I had to guess different authors wrote different subsections). When referencing genes use the mouse or human (or other species) italicized according to current established rules, that way the cited paper has the necessary context without having to look. When discussing proteins, the name should be all caps (regardless of species) and shouldn’t use hyphens or Greek letters, by convention. Proper use allows for easy distinction between references that speak to expression changes, but maybe not functional changes, etc. A minor point, but rules are rules for a reason.
Figure 1, if the author’s permission was provided to use an altered version of that figure, the legend should state so.
Section 2 starts off ABPs in Endometriosis (with just a sentence) then subsections from each class.
It would read nicely if there was a very brief topic sentence or two discussing the rationale for the breadth of ABP to be discussed and if the sections are organized by putative relevance. There could be data about how many ABPs exist in rodents, humans, etc. and what percentage are thought to be relevant based on current literature. This doesn’t have to be extensive, just something to prime the pump rather than starting off Target 1, Target 2, without a preamble. The author’s own section 2.12 kind of does this for the “other” family, so it would be good to mirror a brief discussion here like that one.
With respect to Section 2.12, it is not clear whether Angiogenins are part of the “other ABP” or a third thing to consider? Maybe converting section 2.12 to Section 3. Then subheadings 3.1 for Plastins, 3.2 for WASP, and 3.3 for Angiogenin might make it more clear.
The table might have too many words to make it easy to peruse, but it’s also quite comprehensive and useful, so I’ll defer to the editors on formatting that.
Author Response
Dear Reviewer,
Please see the attachment
Best regards,
Magdalena Izdebska

Reviewer 2 Report
Comments and Suggestions for Authors
General
The ms is a review on development of actin-binding proteins (ABP) in endometriosis and their involvement(s) in the pathological process; with hypothesis on possible targets for therapeutic options in the endometriosis.
About the paper
- Strength. An attempt to develop involvement of cytoskeletal proteins in endometriosis and a targeted medical treatment.
- Limits and insufficiencies. (i) The identifications of medicines suffer from lacunae and imprecisions; (ii) some strategies targeting cytoskeletal proteins have been cited without conclusion on the pathological process involved; (iii) the place of inflammatory ligands as activation mediators is lacking; and in general (iv) too imprecise documentation.
Expression
- Figure 1. Stabulization, constraction do not exist,
- Page 3, line 88. Please use a more appropriate wording for ‘to rise with the distance’,
- Page 5, lines 177, 208; page 8, lines 344, 380. Please use crosslink, more appropriate than link when are referred protein-protein interactions,
- Page 11, line 461. Please write Fusobacterium nucleatum using italic fonts,
- Table 1, row calponin. Please write its level is the highest …
- Please avoid imprecise wording; reveal – incl revealing, reveals, revealed, seem – incl seemed -,
- Please identify all the abbreviations used.
Title
Identifying therapeutic targets is inadequate.
The word therapeutic targets a medicine application or a surgery. Therapeutic in the title makes the reader expecting what are the advantages of the described strategies, ie. by anticipation for the future, over the current options.
Comments
Below the authors are invited to consider some examples of imprecise assumptions.
Section 2.1. α-actinin
- Page 3, lines 89-90. The authors considered that a higher level of α-actinin-1 expression highlights the potential of α-actinin-1 as a therapeutic target in endometriosis, and they are invited to show how a high level of α-actinin-1 is pathogenic.
Section 2.2. Calponin
- Page 4, line 131-134. The authors claim that calponin could serve as a biomarker for early-stage endometriosis, and with its role in stromal dynamics makes it a potential target for therapeutic strategies.
This is highly speculative. The authors are recommended to more deeply develop the medicine and its concept.
Section 2.3. Cofilin-1 (CFL1)
- Page 4, line 160. Silencing LIMK1 is presented to significantly reduce estradiol-induced CFL1 phosphorylation. Is the medicine targeted to one or more kinases ? Is this trial conclusive for endometriosis pathogenic process ?
Section 2.4. Ezrin-radixin-moesin (ERM) family
- Pages 5, line 191. The increase in ezrin and phosphoezrin levels from controls to eutopic and ectopic samples is presented as correlating with enhanced cellular invasiveness. The authors are invited to complement this information regarding pathological process; eg. eutopic versus ectopic, severe deep endometriosis versus less severe adenomyosis, tissue extension.
- Line 195. It is question of inhibition of ezrin phosphorylation. What is the kinase involved? which medicine?
Section 2.6. Myosins and caldesmon
- Page 7, line 296. Contractions are presented as activator of nociceptors in the peritoneum. Because it is directly involved in pain development the authors are recommended to introduce one/several reference(s).
Section 2.7. Talin
- Page 9, lines 368-372. The authors are recommended to provide arguments for a mechanistic process of a modulation of Treg function and differentiation by talin-1.
Section 2.8. Tensins.
- Page 9, line 401. If tensin-1 could be considered as a biomarker for monitoring, which argument may serve as a therapeutic target ?
Section 4. Future perspective
- In order to improve the ms, efforts must be developed to put forward the ABP hypothesis within some other participants in the endometriosis process, in particular the inflammatory triggers.
Conclusion of the referee
- A too abundant information, however insufficient regarding therapeutic options, gives this document a busy appearance and can distract the reader,
- The authors are highly recommended to convince the reader on the position of ABP-targeted medicines for endometriosis,
- This ms suffers from insufficiencies to meet the requirements of the journal.

The authors are recommended to develop with the particular attention the right word for reporting the observations.
Author Response

(The authors gave the same response as above.)

Reviewer 3 Report
Comments and Suggestions for Authors
In this review the authors describe the actin-binding proteins (ABPs) that, by mediating their connection with actin, also modulate its interactions with the cytoskeleton. Below they have listed all the ABPs implicated in this association, reporting their different expressions and regulations (phosphorylation) in relation to Endometriosis.
I found these data well described and analyzed, clearly and logically explained and organized in Figure and Table.
I consider this review very interesting and explanatory and useful in understanding the particularities related to ENDO.
The authors should provide a brief description of how their research was performed (i.e. key words, database, etc.)
Minor:
Line 13: migration (type error)
Figure 1 F-actin STABILIZTION (type error)
Author Response

(The authors gave the same response as above.)

Round 2
Reviewer 2 Report
Comments and Suggestions for Authors
General
The authors answered all the comments and implemented the recommendations of the referee; the quality of the ms is improved.
The referee appreciates the version 2 title.
A final request
Page 1, line 20. In accordance to the modified title, the authors are invited to write ‘we explore the putative potential…’
I appreciate the text for future perspective section.
